# Evaluation of the Relationship between Osteoporosis Parameters in Plain Hip Radiography and DXA Results in 156 Patients at a Single Center in Turkey

**DOI:** 10.3390/diagnostics13152519

**Published:** 2023-07-28

**Authors:** Gokhan Ilyas, Fikri Burak Ipci

**Affiliations:** Department of Orthopedics and Traumatology, Faculty of Medicine, Usak University, Usak 64000, Turkey; fikriburakipci@windowslive.com

**Keywords:** DXA, osteoporosis, Singh, Dorr, cortical thickness, canal-to-calcar ratio

## Abstract

Background: The aim of the current study is to determine the relationship between osteoporosis findings in plain X-ray and dual-energy X-ray absorptiometry (DXA) measurement results and to create an alternative diagnostic method for osteoporosis without DXA measurement when necessary. Methods: DXA values and hip radiographs of 156 patients were retrospectively analyzed. Singh index (SI), Dorr index (DI), cortical thickness index (CTI), and canal-to-calcar ratio (CCR) measurements from both plain hip radiographs were determined by two observers. The correlation of the DXA parameters (hip total T-score, femoral neck T-score, hip total Z-score, hip total bone mineral density [BMD], and femoral neck BMD) and osteoporosis markers on plain hip radiography (SI, DI, CTI, and CCR) was calculated. In addition, patients were evaluated by dividing them into three groups according to the level of their T-scores (normal, osteopenia, and osteoporosis). In addition, cut-off values were calculated for CTI and CCR. Results: The mean age was 68.27 ± 8.27 (50–85) years. There was a strong correlation between hip total T-score values and SI, DI, and CTI (r = 0.683, −0.667, and 0.632, respectively), and a moderate correlation (r = −0.495) with CCR. When both hips were compared, there were strong correlations between radiographic parameters (r = 0.942 for SI, 0.858 for DI, 0.957 for CTI, and 0.938 for CCR, all with *p* < 0.001). When patients divided into three groups according to the T-score level were compared in terms of SI, DI, CTI, and CCR, it was found to be directly related to osteoporosis level (all *p* < 0.001). In the differentiation of osteopenia and osteoporosis, the cut-off values were 50.4 degrees for CTI and 60.3 degrees for CCR. Conclusion: Good correlations between hip total T-score values and SI, DI, CTI, and CCR (r = 0.683, −0.667, 0.632, and −0.495, respectively) indicate that the presence of osteoporosis can be detected by hip radiography findings without DXA.

## 1. Introduction

Osteoporosis is a disease characterized by the deterioration of bone tissue and microarchitecture and a progressive decrease in the mass of the bone [1]. The gold standard method for diagnosing osteoporosis is the estimation of hip and lumbar spine bone mineral density (BMD) by dual-energy X-ray absorptiometry (DXA) [2]. According to the WHO guidelines, a T-score value of 1 to 2.5 standard deviations below the mean bone mass of young adults is defined as osteopenia, while a T-score of less than 2.5 standard deviations is defined as osteoporosis. Although DXA is the gold standard in the diagnosis of osteoporosis, it cannot be used widely in developing countries due to its limited availability [3,4]. Quantitative ultrasonography has been developed as an alternative to osteoporosis screening but has been deemed insufficient to replace DXA [4]. In the diagnosis of osteoporosis, the Fracture Risk Assessment Tool (FRAX), Garvan Fracture Risk Calculator, QFracture algorithm, and Osteoporosis Self-Assessment Tool can also be used [4,5]. Recently, artificial intelligence (AI) has also been added to these tools [6,7].

There are also osteoporosis measurements that can be detected by direct radiography, such as the Singh index, Dorr index, cortical thickness index, and canal-to-calcar isthmus ratio. The Singh Index (SI) is a method of measuring osteoporosis determined by detecting trabecular patterns in the proximal femur with plain radiographs. According to the Singh index, osteoporosis is divided into six grades (Grades 1–6). As the grade progresses, the level of osteoporosis decreases; grades 1–3 define definitive osteoporosis (Figure 1) [8]. The Dorr index (DI) is divided into three types according to cortical thickness on proximal femur radiographs (Types A-B-C). As the type progresses, the level of osteoporosis increases (Figure 2) [9,10]. The cortical thickness index (CTI) is measured 10 cm distal to the trochanter minor; its decrease indicates that the cortex is thinning, thus increasing osteoporosis (Figure 3) [9,10]. The canal-to-calcar isthmus ratio (CCR) is calculated as the ratio of the canal diameter 10 cm distal to the trochanter minor to the canal diameter at the calcar on plain radiographs. An increase in this ratio indicates an increase in osteoporosis (Figure 4) [9,10].

In the only study in the literature like this study, Sah et al. [11] aimed to reveal the relationship between plain X-ray osteoporosis findings and DXA measurements. It was thought that the case group in this study, which was conducted in 2007, consisted of patients with osteoarthritis, which may adversely affect the results. In addition, the current study was planned considering that the series consisting of 32 patients constituted insufficient power analysis.

The aim of the current study is to determine the relationship between osteoporosis findings in plain X-rays. In addition, we aim not to miss a possible diagnosis of osteoporosis, to refer the patient with suspected osteoporosis detected by X-ray to a center with DXA, or to evaluate the spine and hip results of the patient after the measurement, if available, in our own clinic, and to initiate appropriate treatment when necessary.

## 2. Materials and Methods

### 2.1. Ethics Statement, Source of Patients, Inclusion and Exclusion Criteria

After obtaining the approval of the Institutional Review Board (IRB) of Usak University Faculty of Medicine (2 February 2023, decision number: 57-57-11), 1385 patients who underwent DXA scanning in the last year were retrospectively analyzed for this cross-sectional descriptive study.

Inclusion criteria were female patients aged 50–85 years, having DXA and hip radiographs taken less than three months apart from each other. Exclusion criteria were not being in the specified age group (*n* = 114), male patients (*n* = 102), not having hip radiographs accompanying DXA (*n* = 754), more than three months between DXA and radiographs (*n* = 88), hip radiographs not being of sufficient quality (*n* = 94), and having a previous operation history in any of the hips (*n* = 77) (Figure 5).

### 2.2. Data Collected

The DXA values and hip radiographs of the remaining 156 (11%) patients after exclusion criteria were obtained from Picture Archiving and Communication Systems (PACS) records. While making radiographic measurements, the built-in software Fonet Dicom Viewer v4.1 (Fonet Bilgi Teknolojileri A.Ş., Gölbaşı, Ankara, Turkey) was used. Antero-posterior and lateral radiographs of the hip were used for radiographic measurements. In the supine position, AP radiographs taken with the hip joint at 15° internal rotation, and frog leg lateral radiographs taken with the hip joint at 45° abduction and the knee at 30°–40° flexion were accepted as standard radiographs and were included in the study. SI, DI, CTI, and CCR measurements from hip radiographs were made by two observers separately and inter-observer reliabilities were evaluated. The correlation of the DXA parameters (hip total T-score, femoral neck T-score, hip total Z-score, hip total BMD, and femoral neck BMD) and osteoporosis markers on hip radiography (SI, DI, CTI, and CCR) was calculated separately. Radiographic measurements for both hips were made separately and compared with each other. In addition, patients were divided into two groups according to age (between 50–70 [*n* = 88] and 70–85 [*n* = 68]), and it was evaluated whether age was effective in the correlation.

The patients were divided into three groups according to their T-score values as follows: −1 < normal, −2.5 < osteopenia < −1, and −2.5 > osteoporosis, and comparative evaluations were made between them. The cut-off values between normal bone density and osteopenia according to the ROC curve were calculated by dividing the patients into two groups according to whether the T-score was greater or less than −1. In addition, the cut-off values between osteopenia and osteoporosis were divided into two groups according to whether the T-score was greater or less than −2.5.

### 2.3. Statistical Analysis

SPSS (Statistical Package for the Social Sciences) version 24 (IBM Corp., Armonk, NY, USA) was used for statistical analysis. Fisher’s Exact Test and Chi-Square test were used to compare categorical data. Shapiro–Wilk test was applied to the measurements to be evaluated for normality analysis. The Spearman Coefficient and the Mann–Whitney U analysis were used for the parameters without normal distribution, and the Pearson Coefficient and the independent t-test for those with normal distribution; a *p*-value less than 0.05 was considered significant and an r-value between 0.10 and 0.29 was accepted as small correlation, 0.30 to 0.49 medium, and 0.50 to 1.00 as strong correlation. Fleiss’ kappa (κ) coefficient was used for the inter-observer reliability. A κ value is always between 0 and 1; a higher κ value indicates a better correlation. The κ values were graded as slight (0–0.2), fair (0.21–0.40), moderate (0.41–0.60), substantial (0.61–0.80), and almost perfect (0.81–1). Post hoc analysis was performed using the G*power 3.1.9.7 program (Heinrich-Heine-Universität Düsseldorf, Düsseldorf, Germany).

## 3. Results

The mean age of 156 patients included in the study was 68.27 ± 8.27 (50–85) years. The mean total hip T-score was −1.06 ± 1.07 (−3.5–+1.5), the femoral neck T-score was −0.79 ± 1.23 (−3.3–+3), the total hip Z-score was 0.01 ± 0.97 (−3.1–+2.3), the total hip BMD was 0.88 ± 0.15 (0.54–1.25), and the femoral neck BMD was 0.83 ± 0.14 (0.52–1.26). The average data when they were divided into two groups under 70 years old (*n* = 88) and over (*n* = 68) are summarized in Table 1.

When SI was evaluated, eight patients were grade I, fourteen patients were grade II, forty-seven patients were grade III, sixty patients were grade IV, eighteen patients were grade V, and nine patients were grade VI. There was a strong correlation between SI and hip total T-score and femoral neck T-score (r = 0.683 and r = 0.610, respectively, both with *p* < 0.001). Similarly, there was a strong correlation between SI and hip total Z-score, total femoral BMD, and femoral neck BMD (r = 0.588, r = 0.631, and r = 0.535, respectively, all with *p* < 0.001) (Table 2). There was a strong correlation when comparing both hips (r = 0.942, *p* < 0.001). The patient groups that were divided into three groups (normal (*n* = 72), osteopenia (*n* = 66), and osteoporosis (*n* = 18)) according to their T-score values were compared among themselves to the SI grade. It was observed that the *p*-value was <0.001 in all three comparisons (normal–osteopenia, osteopenia–osteoporosis, and normal–osteoporosis). In the SI inter-observer evaluation, it was seen that the result was substantial agreement (κ = 0.756 (0.712–0.800)).

When evaluated in terms of DI, 77 patients were type A, 57 patients were type B, and 22 patients were type C. Similar to the SI, there was a strong correlation between the DI and hip total T-score, femoral neck T-score, hip total Z-score, total femoral BMD, and femoral neck BMD (r = −0.667, r = −0.574, r = −0.557, r = −0.628, and r = −0.535, respectively, all with *p* < 0.001) (Table 2). There was a strong correlation when comparing both hips (r = 0.858, *p* < 0.001). When the groups separated according to the presence of osteoporosis were compared among themselves (*n*1 = 72, *n*2 = 66, *n*3 = 18) according to the degree of DI, it was found to be statistically highly significant, similar to the SI grade results (normal–osteopenia, osteopenia–osteoporosis, and normal–osteoporosis, all three *p* < 0.001). The DI was found to be in moderate agreement in the inter-observer evaluation (κ = 0.562 [0.519–0.605]).

The mean CTI was 54.5 ± 6.88 (32.2–71.8) percent. While there was a strong correlation between CTI and hip total T-score, femoral neck T-score, hip total Z-score, and total femoral BMD (r = 0.632, r = 0.513, r = 0.509, and r = 0.608, respectively, all with *p* < 0.001), there was a medium correlation with femoral neck BMD (r = −0.459, *p* < 0.001) (Table 3). According to the ROC curve, 68% sensitivity and 68% specificity were found in the distinction of normal bone density and osteopenia at a cut-off value of 55.25 degrees for CTI. In addition, 83% sensitivity and 83% specificity were found in the distinction of osteopenia and osteoporosis at a cut-off value of 50.4 degrees (Table 4). In the evaluation carried out according to the osteoporosis groups, similar results were seen in the CTI as well as in the SI and DI (*n*1 = 72, *n*2 = 66, *n*3 = 18) (all three *p* < 0.001). There was a strong correlation when comparing both hips (r = 0.957, *p* < 0.001).

The mean CCR was 56 ± 9.78% (38–87) percent. There was a medium correlation between CCR and hip total T-score, femoral neck T-score, hip total Z-score, total femoral BMD, and femoral neck BMD (r = −0.495, r = −0.445, r = −0.390, r = −0.434, r = −0.370, respectively, all with *p* < 0.001) (Table 3). When the CCR was evaluated according to the ROC curve, 68% sensitivity and 68% specificity were found in differentiating normal bone density from osteopenia at a cut-off value of 54.9 degrees. Moreover, 78% sensitivity and 77% specificity were found in the distinction between osteopenia and osteoporosis at a cut-off value of 60.3 degrees (Table 4). In the evaluation made according to the osteoporosis groups, similar results were seen in the CCR as well as in the other radiographic parameters (*n*1 = 72, *n*2 = 66, *n*3 = 18) (all three *p* < 0.001). There was a strong correlation when comparing both hips (r = 0.938, *p* < 0.001).

Power (1 − *b*) in a post hoc analysis was calculated as 0.974 (sample size = 156, effect size d = 0.3, and *a* err prob = 0.05). In addition, when comparing two groups according to age, power (1 − *b*) was calculated as 0.914 (sample size (*n*1 = 68–*n*2 = 88), effect size d = 0.5 and *a* err probe = 0.05).

## 4. Discussion

Osteoporosis is considered to be an important cause of morbidity and mortality, especially as it can cause hip fractures [12,13]. In this study, we evaluated the correlation of DXA, which is the most important method in the diagnosis of osteoporosis [2], and radiographic hip findings of osteoporosis. While a strong correlation was found between hip total T-score values and the SI, DI, and CTI (r = 0.683, −0.667, 0.632, respectively), a medium correlation was observed with the CCR (r = −0.495). According to the ROC curve, the probability of osteoporosis is very high in CTI cut-off values below 50.4 degrees and CCR cut-off values above 60.3 degrees.

The main purpose of this study is to enable physicians to pre-diagnose osteoporosis with simple and practical measurements on hip X-rays, to direct the patient to DXA, to initiate treatment if necessary, and to prevent unwanted osteoporotic fractures. Since physicians did not have an examination method to diagnose osteoporosis in the patient, it was seen that an incomplete or wrong diagnosis could be avoided with a simple hip X-ray that can be taken almost anywhere.

Sah et al. measured the SI, DI, CTI, and CCR of 32 hips with osteoarthritis. No statistically significant difference was found between the mean SI grades of the osteoporotic and non-osteoporotic groups (4.2 ± 1.0, 3.9 ± 0.8, respectively) (*p* = 0.406) [11]. They stated that there was no correlation between the SI grade and T-score (*p* = 0.726). In this study, a high level of significance was found in the comparison of SI grades between groups with and without osteoporosis (*p* < 0.001). In addition, there was a strong correlation between the SI and T-score (r = 0.683, *p* < 0.001). In the same study, they stated that the increase in osteoporosis was statistically significant as the DI type progressed from A to C (T-scores −0.538 ± 1.346, −1.817 ± 1.193, −2.244 ± 1.276, respectively) (*p* = 0.028). In the current study, similar to this study, there was a strong correlation between the increase in DI grade and a decrease in T-score (r = −0.667, *p* < 0.001). In the same study, a statistically significant relationship was found between the CTI and T-score (*p* = 0.003–0.004), but no significant relationship was found between the CCR and T-score (*p* = 0.576). The current study showed a strong positive correlation between the CTI and T-score (r = 0.632, *p* < 0.001), and a moderate negative correlation between the CCR and T-score (r = −0.495, *p* < 0.001). Since patients with osteoarthritis were included in this study, it was thought that it might adversely affect some results. In addition, considering the power analysis, it was seen that a higher series was needed. The post hoc analysis result of our study was 0.974.

A study of 60 patients in 2021 looked at the correlation of the CTI with hip and spine T-scores [14]. The CTI was found to be statistically significantly lower in the group determined to be osteoporotic by DXA. In the current study, values compatible with this were determined (r = 0.632, *p* < 0.001), and in addition, the SI, DI, and CCR were evaluated (r = 0.683, −0.667, −0.495, respectively, all three *p* < 0.001). Osteoporosis may affect the decision while determining the treatment method for hip fractures. Since there may be a difference in BMD between the two hips [15], we think that quantitative values of the CTI and CCR can be used instead of DXA in cases with a history of surgery on the other hip (r = 0.957 and 0.938, respectively, both with *p* < 0.001).

Zhu et al. recommended measuring contralateral hip BMD when the total hip T-score is between −1.6 and −2.5 in men and between −1.8 and −2.5 in women, to prevent the diagnosis of osteoporosis from being missed [15]. Considering this study, it was thought that the radiographic findings of osteoporosis in bilateral hips might be different. In addition, Ikegami et al. stated that unilateral femoral DXA measurement is not sufficient to exclude contralateral hip osteoporosis [16]. Unilateral DXA scanning of the hip is performed as a standard in our clinic. Due to this situation, other hip BMD measurements could not be evaluated in this retrospective study. However, the SI, DI, CTI, and CCR were evaluated and compared bilaterally. A strong correlation was detected between radiographic osteoporosis findings of both hips (r = 0.942, 0.858, 0.957, 0.938, respectively, all with *p* < 0.001).

Jang et al. reported that artificial intelligence (AI), which they developed using DXA T-scores and hip radiography, is a useful method (81.2% accuracy, 91.1% sensitivity, and 68.9% specificity) in estimating osteoporosis in a large-series study they designed with two groups, defined as osteoporosis and osteopenia cases [7]. Similarly, Yamamoto et al., in an AI study, evaluated the presence or absence of osteoporosis in hip radiography [6]. They demonstrated that osteoporosis was diagnosed with high accuracy through convolutional neural network models and was further improved by the addition of clinical variables. These AI studies have shown that osteoporosis can be diagnosed directly with a hip X-ray. Although very good results are obtained with artificial intelligence, it seems that it will take a long time before they are available to use in clinical practice. In our study, where we aimed to make a preliminary diagnosis of osteoporosis with a simple examination that can be made on the findings of the direct radiographs, which are still frequently used in daily practice and can be accessed in many health centers, we examined the compatibility of the hip T-score with the osteoporosis findings from the direct X-ray. We examined three groups, osteoporosis, osteopenia, and normal, and tried to estimate the T-score only by direct hip radiography findings. Good correlations between hip total T-score values and direct radiography osteoporosis findings, when DXA could not be reached, we determined the pre-diagnosis of osteoporosis is possible with only hip radiography findings. Referring the patient to DXA in the presence of osteoporosis findings on direct radiographs will prevent osteoporosis from being overlooked. In addition, paying attention to these findings will ensure that osteoporosis can be prevented with an early diagnosis for any patient who applies to the clinic with another hip complaint for which a direct hip radiograph is taken. It is thought that the cut-off values in the differentiation of osteopenia and osteoporosis will be even more effective in reaching the correct diagnosis (50.4 degrees for CTI, and 60.3 degrees for CCR).

Mazhar et al. performed a DI reliability analysis and stated that this resulted in moderate intra-observer and inter-observer agreements and stated that more reliable characterization systems are needed to classify femoral morphology [17]. In another study, inter-examiner reproducibility was found to be κ = 0.32 (fair agreement) in the inexperienced group and κ = 0.52 (moderate agreement) in the experienced group [18]. In the current study, in which two observers were evaluated, the DI showed moderate agreement with inter-observer reliability (k = 0.562 [0.519–0.605]).

Hauschild et al. performed a reliability analysis for the SI and evaluated the mean intra-observer agreement as moderate agreement (κ = 0.43 ± 0.28) and the mean inter-observer agreement as slight agreement (κ = 0.199 ± 0.248) [19]. In the same study, they determined the mean correlation values between SI and trochanteric BMD and T-scores as r = 0.219 ± 0.04 and r = 0.210 ± 0.05, respectively. Inter-observer evaluation for SI in the current study showed substantial agreement (k = 0.756 (0.712–0.800)).

In a study, Taafe et al. showed a weak correlation between hip BMD as measured by DXA and middle femoral bone mineral content measured by computed tomography and volumetric bone mineral density (vBMD) [20]. In the current study, there was a good level of correlation between hip total T-score values and SI, DI, CTI, and CCR measurements (r = 0.683, −0.667, 0.632, −0.495, respectively). Although a 90% reduction in radiation is reported in low-dose CT compared to conventional CT [21], it should be kept in mind that there is more radiation exposure than in plain radiographs.

In addition, there are many studies comparing cortical thickness and DXA measurements for different bone localizations [22,23,24,25,26]. This explains how much importance should be given to BMD.

This study has some limitations. The most important limitation of the study is that trabecular and cortical vBMD and vitamin D levels could not be measured due to its retrospective nature. The fact that every hip radiograph taken was not in the ideal position was determined as an additional limitation. It is possible to prevent these situations by planning prospectively.

## 5. Conclusions

In cases where DXA, which is the most important evaluation method in the diagnosis of osteoporosis, cannot be reached, it has been observed that the findings of hip direct radiography can be supportive. In addition, it was determined that DXA should be used especially in cases where direct radiography findings support osteoporosis. The good correlations between hip total T-score values and the SI, DI, CTI, and CCR (r = 0.683, −0.667, 0.632, and −0.495, respectively) support this conclusion. CTI cut-off values below 50.4 degrees and CCR cut-off values above 60.3 degrees were found to correspond to the probability of osteoporosis.

## Figures and Tables

**Figure 1 diagnostics-13-02519-f001:**
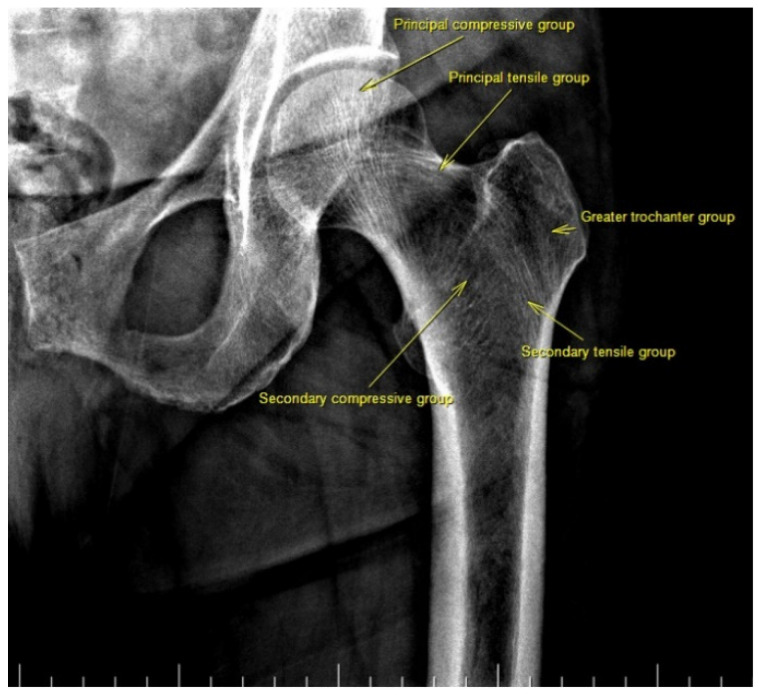
Singh index consists of six grades. In grade six, all trabecular patterns are visible and of normal thickness. In grade five, Ward’s triangle is prominent, and the principal tensile and principal compressive trabeculae are easily visible. In grade four, the principal tensile trabeculae are continuous but thinned. In grade three, loss of continuity is accompanied by thinning of the principle tensile trabeculae. In grade two, except for the principal compression trabeculae, the trabeculae are almost invisible. In grade one, principal compressive trabeculae are visible as thin. Antero-posterior hip radiograph of a patient with Singh grade 6 is shown.

**Figure 2 diagnostics-13-02519-f002:**
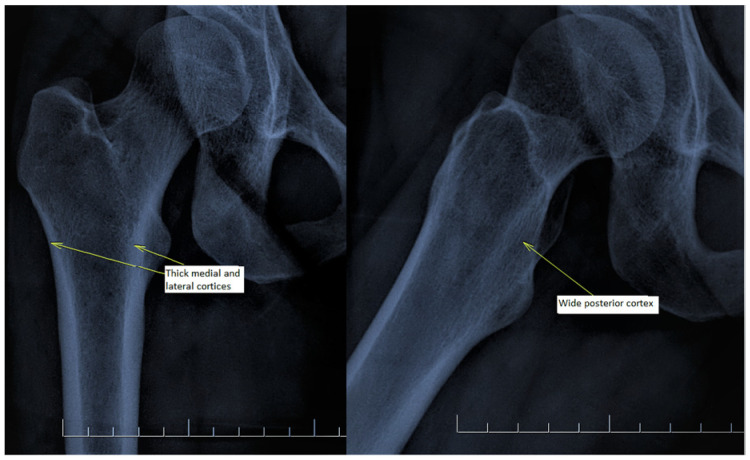
Dorr index consists of three types. Type A: thick medial and lateral cortices on hip antero-posterior X-ray and a wide posterior cortex on lateral X-ray. Type B: medial cortex and proximal part of the posterior cortex are thinned. Type C: medial and posterior cortices are lost. Antero-posterior and lateral radiographs of a patient with Dorr Type A are shown.

**Figure 3 diagnostics-13-02519-f003:**
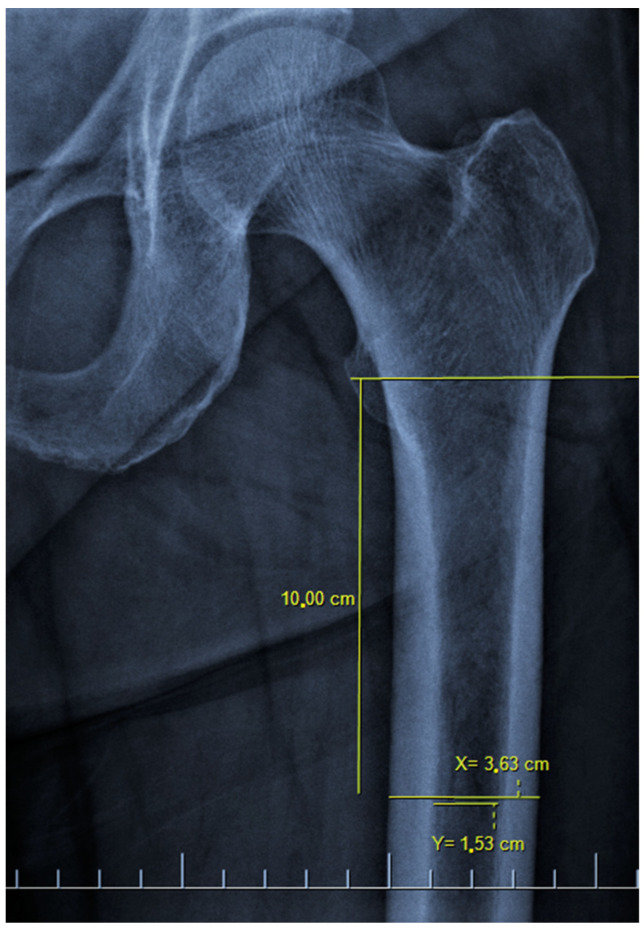
The cortical thickness index is measured 10 cm distal to the mid-lesser trochanteric line ((X − Y)/X). It is defined as the ratio of the difference between the diameter of the femoral diaphysis (X) and the diameter of the intramedullary canal (Y) to the diameter of the femoral diaphysis. It was measured as 0.58 ((36.3 − 15.3)/36.3) on the X-ray of the patient.

**Figure 4 diagnostics-13-02519-f004:**
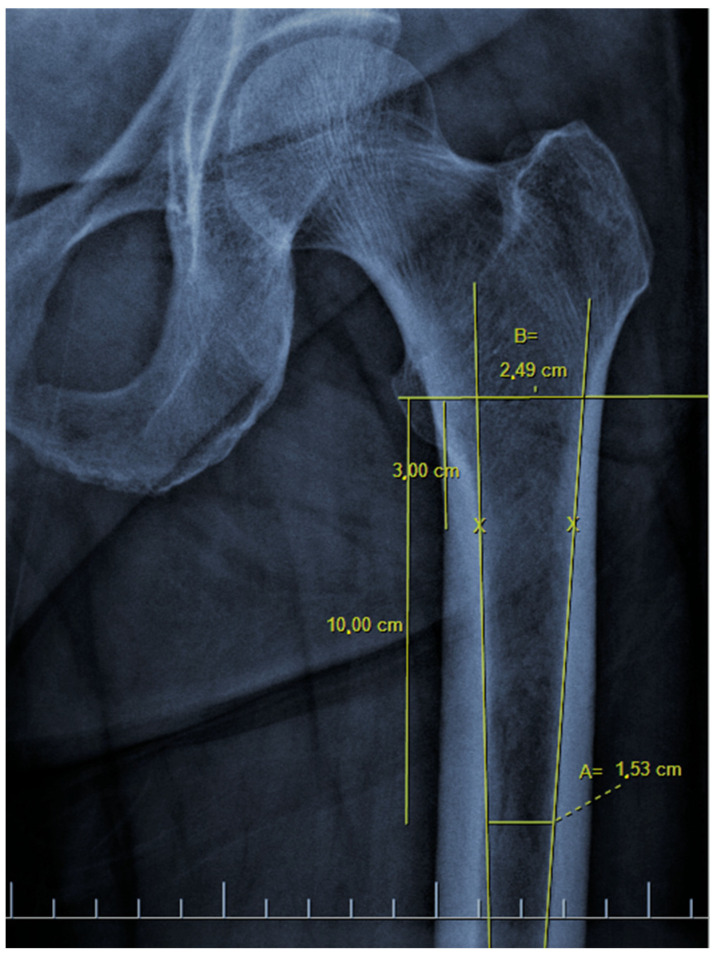
The canal-to-calcar ratio (A/B). The diameter of the medulla between the endosteal points 10 cm distal and the mid-lesser trochanteric line is measured (A). Endosteal points located 3 cm distal to the mid-lesser trochanteric line are determined (x). The lines connecting the distal endosteal points with the proximal endosteal points are extended to the mid-lesser trochanteric line and the distance between the lines at this level is measured (B). It was measured as 0.61 (15.3/24.9) on the x-ray of the patient.

**Figure 5 diagnostics-13-02519-f005:**
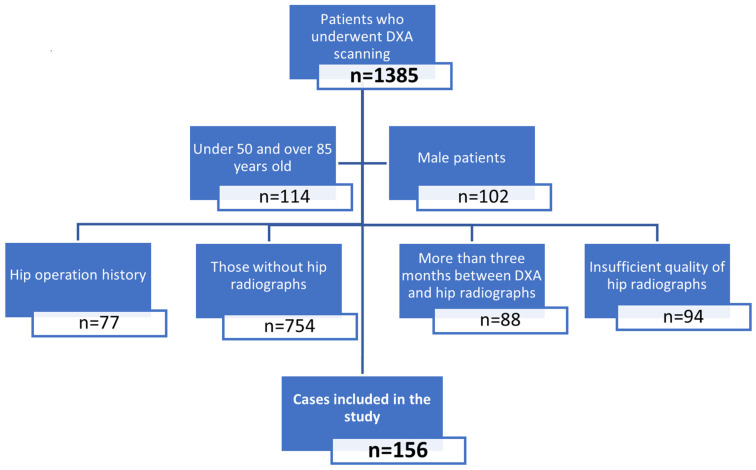
Flow chart of cases.

**Table 1 diagnostics-13-02519-t001:** DXA values by age groups.

	Over 70 Years Old (*n* = 68)	Between 50 and 70 Years (*n* = 88)	*p*-Value *
Total hip T-score mean ± SD (range)	−1.49 ± 0.96 (−3.40–+0.30)	−0.73 ± 1.04 (−3.50–+1.50)	<0.001
Femoral neck T-score mean ± SD (range)	−1.24 ± 1.07 (−3.20–+1.40)	−0.44 ± 1.24 (−3.3–+1.60)	<0.001
Total hip Z-score mean ± SD (range)	−0.17 ± 0.90 (−2.0–+1.60)	0.12 ± 1.01 (−3.10–+2.30)	0.065
Total hip BMD mean ± SD (range)	0.82 ± 0.14 (0.54–1.07)	0.92 ± 0.14 (0.62–1.25)	<0.001
Femoral neck BMD mean ± SD (range)	0.78 ± 0.12 (0.52–1.06)	0.86 ± 0.16 (0.50–1.26)	<0.001

DXA: Dual-energy X-ray absorptiometry, SD: Standard deviation, BMD: Bone mineral density. * The independent *t*-test was used.

**Table 2 diagnostics-13-02519-t002:** The Correlation between the Hip DXA Values and Singh and Dorr Indices.

	Singh Index			Dorr Index		
	r-Value *	Interpr.	*p*-Value	r-Value *	Interpr.	*p*-Value
Total hip T-score	0.683	strong cr.	<0.001	−0.667	strong cr.	<0.001
Femoral neck T-score	0.610	strong cr.	<0.001	−0.574	strong cr.	<0.001
Total hip Z-score	0.588	strong cr.	<0.001	−0.557	strong cr.	<0.001
Total hip BMD	0.631	strong cr.	<0.001	−0.628	strong cr.	<0.001
Femoral neck BMD	0.535	strong cr.	<0.001	−0.535	strong cr.	<0.001

DXA: Dual-energy X-ray absorptiometry, BMD: Bone mineral density, cr: correlation, interpr: interpretation. * The Pearson correlation coefficient was used.

**Table 3 diagnostics-13-02519-t003:** The Correlation between the Hip DXA Values and CTI and CCR.

	CTI			CCR		
	r-Value *	Interpr.	*p*-Value	r-Value *	Interpr.	*p*-Value
Total hip T-score	0.632	strong cr.	<0.001	−0.495	medium cr.	<0.001
Femoral neck T-score	0.513	strong cr.	<0.001	−0.445	medium cr.	<0.001
Total hip Z-score	0.509	strong cr.	<0.001	−0.390	medium cr.	<0.001
Total hip BMD	0.608	strong cr.	<0.001	−0.434	medium cr.	<0.001
Femoral neck BMD	0.459	medium cr.	<0.001	−0.370	medium cr.	<0.001

DXA: Dual-energy X-ray absorptiometry, CTI: Cortical thickness index, CCR: Canal-to-calcar. isthmus ratio, BMD: Bone mineral density, cr: correlation, interpr: interpretation. * The Pearson correlation coefficient was used.

**Table 4 diagnostics-13-02519-t004:** Cut-off values * for CTI and CCR.

	Between Normal Bone Density and Osteopenia (Degrees)	Between Osteopenia and Osteoporosis (Degrees)
CTI (sensitivity%–specificity%)	55.25 (68–68)	50.4 (83–83)
CCR (sensitivity%–specificity%)	54.9 (68–68)	60.3 (78–77)

CTI: Cortical thickness index, CCR: Canal-to-calcar isthmus ratio. * According to the ROC curve.

## Data Availability

Data in the current study can be obtained from the corresponding author.

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
