# Peer review of "Evaluation of the Relationship between Osteoporosis Parameters in Plain Hip Radiography and DXA Results in 156 Patients at a Single Center in Turkey"

_diagnostics, 2023, doi:10.3390/diagnostics13152519_

Round 1
Reviewer 1 Report
Dear Authors,
I hope this email finds you well. I have reviewed the manuscript titled "Evaluation of the Relationship Between Osteoporosis Parameters on Plain Hip Radiography and DXA Results in 156 Patients" and found it to be intriguing. However, I have a few questions that I believe the authors should address:
1. How can the study findings support the use of hip radiography findings as an alternative diagnostic method for osteoporosis without relying on DXA measurements?
2. Given that plain hip radiography provides 2D images, how did the authors address the measurement of hip parameters in all directions? Was there any specific approach taken to ensure consistent patient positioning during radiography, considering the importance of uniformity in measuring parameters like thickness?
3. While hip radiography shows promise in improving the diagnosis of osteoporosis as a supplementary diagnostic tool, I think that it cannot fully replace DXA. Could the conclusion be slightly modified to reflect this aspect?
4. Can osteoporosis be reliably identified solely through hip radiography, considering that DXA typically measures the femur and vertebrae to assess osteoporosis?
5. Were there established standards for the thickness of medial and lateral cortices, and if so, how did the authors compare their results against these standardized criteria and other relevant parameters? Which specific criteria were utilized for comparison?
Kind regards,
Author Response
Thanks for your valuable comments. I will try to explain your reviews item by item. Changes made were shown in red font.
- In this study, we tried to state that DXA is required in cases where it is necessary as a result of direct radiography findings, rather than that DXA measurement is not required. Since DXA measurement cannot be performed in every medical center or clinic, it was aimed to predict that the patient has osteoporosis with direct radiograph findings and refer the patient to the DXA center when necessary. In addition, when the patient applied to the outpatient clinic with another hip complaint, it was determined that the patient had osteoporosis on the x-ray scanned randomly, and the patient was directed to the DXA measurement, which requires a possible treatment, and it is aimed not to miss the osteoporosis. The existence of the determined cut-off values also supports this situation. The conclusion part has been rearranged accordingly.
- Patients with standard hip AP and lateral radiographs were accepted for the study. It was added in the material section that the patients were scanned with standard hip radiographs and the measurements were made.
- The conclusion part has been reorganized accordingly.
- Our aim in this study is not to miss a possible diagnosis of osteoporosis, to refer the patient with suspected osteoporosis on X-ray to a center with DXA, or to evaluate the spine and hip results of the patient after the measurement, if available in our clinic, and to initiate appropriate treatment when necessary. Such an arrangement has been made in the last paragraph of the Introduction section.
- "Three distinct patterns of shape and bone structure of the femur were
qualitatively identified from roentgenographs between the metaphysis and the diaphyseal isthmus." as stated in Dorr's original article. Examined as implemented in the original study, no net value in "mm" was used.
Reviewer 2 Report
Discussion with the work of Jang and Yamamoto that pioneered the use of AI in X-ray image analysis is insufficient. Please clearly indicate how your results differ and how they can help an orthopedist who watches hip X-rays in everyday practice. It's about the more practical results of your research.
Discussion needs to be more "meaty". For many years it was believed that the indexes tested in the work are not very accurate diagnostics of osteoporosis, but in clinical practice many orthopedists diagnose osteoporosis in this way and refer their patients to further DEXA tests. The work should have more practical implications for physicians
Author Response
Thank you for your constructive review.
I've more appropriately expanded the discussion of the work done for AI, adding both studies to talk about how it can help my colleagues.
Additions have been made to the discussion ce conclusion sections.
Changes made were shown in red font.